# Kinetics of Prostate-Specific Antigen after Carbon Ion Radiotherapy for Prostate Cancer

**DOI:** 10.3390/cancers12030589

**Published:** 2020-03-04

**Authors:** Narisa Dewi Maulany Darwis, Takahiro Oike, Hidemasa Kawamura, Masahiro Kawahara, Nobuteru Kubo, Hiro Sato, Yuhei Miyasaka, Hiroyuki Katoh, Hitoshi Ishikawa, Hiroshi Matsui, Yoshiyuki Miyazawa, Kazuto Ito, Kazuhiro Suzuki, Soehartati Gondhowiardjo, Takashi Nakano, Tatsuya Ohno

**Affiliations:** 1Department of Radiation Oncology, Gunma University Graduate School of Medicine, 3-39-22, Showa-machi, Maebashi, Gunma 371-8511, Japan; narisajimmy@gmail.com (N.D.M.D.); kawa@gunma-u.ac.jp (H.K.); kubo0330@gmail.com (N.K.); h-sato0808@hotmail.co.jp (H.S.); myuhei.76@gmail.com (Y.M.); hkatoh@kcch.jp (H.K.); hishikawa@pmrc.tsukuba.ac.jp (H.I.); tnakano@gunma-u.ac.jp (T.N.);; 2Department of Radiation Oncology, Faculty of Medicine Universitas Indonesia – dr. Cipto Mangunkusumo Hospital, Jl. P. Diponegoro no. 71, Jakarta 10430, Indonesia; gondhow@gmail.com; 3Gunma University Heavy Ion Medical Center, 3-39-22, Showa-machi, Maebashi, Gunma 371-8511, Japan; matsuih@gunma-u.ac.jp (H.M.); kazu@gunma-u.ac.jp (K.S.); 4Department of Radiation Oncology, Saku Central Hospital Advanced Care Center, 3400-28, Nakagomi, Saku, Nagano 385-0051, Japan; masahr.kawa@gmail.com; 5Department of Radiation Oncology, Kanagawa Cancer Center, 2-3-2, Nakao, Asahi-ku, Yokohama, Kanagawa 241-8515, Japan; 6Department of Radiation Oncology, Faculty of Medicine, University of Tsukuba, 1-1-1, Tennodai, Tsukuba, Ibaraki 305-8575, Japan; 7Department of Urology, Gunma University Graduate School of Medicine, 3-39-22, Showa-machi, Maebashi, Gunma 371-8511, Japan; miya.yoshi@gunma-u.ac.jp (Y.M.); kzito@gunma-u.ac.jp (K.I.); 8Institute for Preventive Medicine, Kurosawa Hospital, 187, Yanaka-machi, Takasaki, Gunma 370-1203, Japan; 9National Institutes for Quantum and Radiological Science and Technology, 4-9-1, Anagawa, Inage-ku, Chiba, Chiba 263-8555, Japan

**Keywords:** prostate cancer, carbon ion radiotherapy, prostate-specific antigen (PSA), PSA bounce

## Abstract

This study aimed to first elucidate prostate-specific antigen (PSA) kinetics in prostate cancer patients treated with carbon ion radiotherapy (CIRT). From 2010 to 2015, 131 patients with prostate adenocarcinoma treated with CIRT (57.6 Gy relative biological effectiveness (RBE) in 16 fractions) alone were recruited. PSA was measured at 1, 2, 3, 6, 9, 12, 15, 18, 21, 24, 30, 36, 42, 48, 54, and 60 months post-CIRT. PSA bounce was defined as PSA increase over a cutoff followed by spontaneous decrease to or below the pre-bounce nadir. PSA failure was determined using the Phoenix criteria (nadir + 2.0 ng/mL). As a result, non-failure-associated temporary increase in PSA exhibited two distinct patterns, namely a classical bounce and a surge at one month. PSA bounce of ≥0.2 ng/mL was observed in 55.7% of the patients. Bounce amplitude was <2.0 ng/mL in 97.6% of cases. Bounce occurred significantly earlier than PSA failure. Younger age was a significant predictor of bounce occurrence. Bounce positivity was a significant predictor of favorable 5-year PSA failure-free survival. Meanwhile, a PSA surge of ≥0.2 ng/mL was observed in 67.9% of patients. Surge amplitude was significantly larger than bounce amplitude. Larger prostate volume was a significant predictor of PSA surge occurrence. PSA surge positivity did not significantly predict PSA failure. In summary, PSA bounce was distinguishable from PSA failure with regard to timing of occurrence and amplitude (earlier and lower for bounce, respectively). These data are useful for post-CIRT surveillance of prostate cancer patients.

## 1. Introduction

Radiotherapy plays a pivotal role in the management of prostate cancer [1]. Carbon ion radiotherapy (CIRT) has potential as a definitive radiotherapy modality for localized prostate cancer [2]. Prostate-specific antigen (PSA) is the gold standard biomarker for post-treatment surveillance of prostate cancer patients [3,4]. In curative cases, PSA levels decrease gradually over >5 years post-radiotherapy and eventually reach a nadir [5,6]. Nevertheless, PSA levels fluctuate and temporarily increase in some patients, a phenomenon called the PSA bounce [7]. This bounce causes severe anxiety for prostate cancer patients and clinicians [8], and may even lead to unnecessary salvage treatment in cases that meet the definition of PSA failure [9]. The bounce has been extensively studied in the context of external beam radiotherapy (EBRT) using photons [10], stereotactic body radiotherapy (SBRT) [11], and high- and low-dose-rate brachytherapy [6,12]. However, PSA bounce after CIRT has not been examined. Hence, we analyzed post-treatment PSA kinetics in prostate cancer patients treated with CIRT without androgen deprivation therapy (ADT).

## 2. Results

The study included 131 patients (Table 1). Median follow-up was 60 months (range, 39–60 months). PSA data were obtained at 98.7% and 93.6% of planned time points within 3 and 5 years post-CIRT, respectively, indicating high follow-up compliance.

Post-CIRT PSA kinetics for all patients are summarized in Figure 1A,B. Eight patients experienced PSA failure (Figure 1A). In PSA failure-free patients (Figure 1B), the patterns of temporary increase in PSA could be classified into two distinct groups in terms of the timing of occurrence and amplitude, namely one that is classically known as the PSA bounce (Figure 1C,E) and the other being a sharp rise in PSA at one month post-CIRT (Figure 1D,E). The latter phenomenon has not been reported previously; therefore, we refer to this hereafter as the PSA surge.

Using a commonly employed cutoff of 0.2 ng/mL [9,11,12,13,14], a PSA bounce was observed in 55.7% (73/131) of patients. The PSA bounce occurred significantly earlier than increases associated with PSA failure (15.0 ± 11.4 vs. 51.0 ± 7.6 months, *p* < 0.0001) (Figure 2). Bounce amplitude was within 2.0 ng/mL in 97.6% of cases (Figure 2). Univariate and multivariate analysis revealed that younger age was associated with a higher likelihood of bounce (Table 2).

PSA surge of ≥0.2 ng/mL was observed in 67.9% (89/131) of patients. In all cases, the surge peaked out quickly and recovered to or below the pre-surge level within 2 months (Figure 1A and Figure 2). Interestingly, the amplitude of PSA surge compared with that for PSA bounce (3.39 ± 2.78 ng/mL vs. 0.79 ± 1.02 ng/mL, *p* < 0.0001). Univariate and multivariate analysis showed that greater prostate volume was the predictor for higher likelihood for surge (Table 2).

A PSA bounce was a significant predictor of favorable PSA failure-free survival when using a cutoff of 0.2 ng/mL, but not higher values (Table 3). With a cutoff of 0.2 ng/mL, the 5-year PSA failure-free survival rates for bounce-positive and -negative patients were 100% and 81.5%, respectively (*p* <0.001) (Figure 3A). By contrast, PSA surge did not have predictive significance for PSA failure, regardless of the cutoff value (≥0.2 to 2.0 ng/mL) (Table 3, Figure 3B).

## 3. Discussion

This was the first report of post-treatment PSA kinetics in prostate cancer patients treated with CIRT. Data quality was high, as indicated by the standardized treatment, high follow-up compliance, and the fact that PSA measurements were all made at a single institution. PSA bounce was distinguishable from PSA failure, in terms of the timing of occurrence and amplitude (earlier and lower, respectively, for the bounce). The data may provide a clinical insight on post-treatment surveillance of CIRT; PSA bounce should be carefully ruled out from failure especially within 36 months post-CIRT to prevent unnecessary salvage treatment even when the amplitude exceeds 2.0 ng/mL, whereas a continuous increase in PSA observed after 36 months is more probable for failure than bounce. We also discovered a novel PSA kinetics, that is, surge, that warrants further investigation.

The incidence of PSA bounce varies widely among studies, where the various cutoffs employed make inter-study comparison difficult. When focused on the studies that employed the same cutoff as in our study (i.e., ≥0.2 ng/mL), the incidence is approximately 25%, 35%, 40%, and 40% for EBRT, SBRT, and high- and low-dose-rate brachytherapy, respectively (Appendix A). In this study, the incidence of PSA bounce for CIRT was 55.7%, which was higher than that reported for other radiotherapy modalities, warranting future validation.

High prevalence of PSA bounce among younger patients was consistent with previous reports on SBRT [15,16] and brachytherapy [17]. The mechanisms underlying PSA bounce and its association with age have not been fully elucidated. However, Yamamoto et al. demonstrated an increase in tumor-infiltrative CD3 and cytotoxic CD8 cells in bounce-positive patients [18], seemingly explaining the higher incidence of PSA bounce in younger (i.e., more immunocompetent) patients.

Predictive significance of PSA bounce with a cutoff of 0.2 ng/mL on favorable PSA failure-free survival was consistent with previous reports on EBRT [19] and brachytherapy [17]. In our study, follow-up was limited to 5 years. Therefore, a longer follow-up will clarify this issue further.

PSA surge has not been reported previously for any radiotherapy modality. However, we cannot conclude at this stage that PSA surge is a CIRT-specific phenomenon because most previous studies analyzing PSA kinetics after non-CIRT radiotherapies lack the one-month post-treatment time point. Differences between PSA surge and bounce in predictive clinical factors and in predictive ability for PSA failure indicate that they are physiologically distinct. Predictive significance of greater prostate volume on higher incidence and no predictive significance on PSA failure indicate that PSA surge may derive from the normal prostate tissues irradiated with carbon ions. Further research is needed to elucidate this issue.

## 4. Materials and Methods

### 4.1. Patient Eligibility

This was a prospective observational study including patients with newly diagnosed, histologically confirmed prostate adenocarcinoma who met the following criteria: (i) treated with CIRT at Gunma University Heavy Ion Medical Center, between July 2010 and July 2015; (ii) staged as T1–T3N0M0 according to the International Union Against Cancer TNM classification (2002); (iii) no neoadjuvant, adjuvant, or concurrent ADT; and (iv) followed up at least 3 years post-CIRT. This study was approved by the institutional ethical review board (Ethical Review Committee of Gunma University Hospital. The approval code: #693. The date of approval: 26 May 2010). Written informed consent was obtained from all patients.

### 4.2. Carbon Ion Radiotherapy

CIRT was performed as described previously [20]. Briefly, the patients’ feet were set in a customized cradle (Moldcare; Alocare, Tokyo, Japan), and the pelvis was immobilized using a low-temperature thermoplastic sheet (Shellfitter; Kuraray, Co., Ltd., Osaka, Japan). Sterilized saline (100 mL) was inserted into the bladder, and the rectum was emptied using an enema. Computed tomography (CT) images were obtained at 2 mm thickness. Treatment planning was performed using Xio-N software (Elekta, Stockholm, Sweden and Mitsubishi Electric, Tokyo, Japan) on the CT images fused with magnetic resonance images. Clinical target volume (CTV) included the prostate and the proximal seminal vesicles (SVs). Planning target volume (PTV)1 was created by adding the anterior and lateral margins of 10 mm, cranial and caudal margins of 6 mm, and a posterior margin of 5 mm to the CTV, with lateral margins to the SVs of 3 mm. PTV2 was created by modifying PTV1, where the posterior edge was set in front of the anterior wall of the rectum. Each field was designed using spread-out Bragg peak, multi-leaf collimators, and custom-made compensation bolus. Carbon ion irradiation consisted of 57.6 Gy (relative biological effectiveness (RBE)) in 16 fractions (4 fractions per week). The first nine sessions targeted PTV1, followed by seven sessions that targeted PTV2. One irradiation field was used for each session; one anterior field and lateral opposing fields for PTV1, and lateral opposing fields for PTV2. Sterilized saline (100 mL) was inserted into the bladder for the session using the anterior field. Patient positioning was three-dimensionally corrected using bone-matching systems.

### 4.3. PSA Assessment

CIRT was initiated on day 0. PSA was measured at 1, 2, 3, 6, 9, 12, 15, 18, 21, 24, 30, 36, 42, 48, 54, and 60 months post-CIRT at Gunma University Hospital. PSA bounce was defined as an increase in PSA over a cutoff followed by a spontaneous decrease to or below the pre-bounce nadir [17]. PSA failure was determined based on the Phoenix criteria (nadir + 2.0 ng/mL) [21]. A temporary PSA increase of ≥2.0 ng/mL was not classified as failure [22].

### 4.4. Statistical Analysis

Association of clinical variables with PSA kinetics was assessed by logistic regression [17]. Association of PSA kinetics with PSA failure-free survival was assessed by the Kaplan–Meier method with log-rank comparison [17,19]. Differences in continuous variables between the two groups were evaluated using the Mann–Whitney *U* test. Statistical analyses were performed using Stata/MP 13 (Stata Corp, College Station, TX, USA) or Prism8 (GraphPad Software, San Diego, CA, USA). *p* < 0.05 was considered statistically significant.

## 5. Conclusions

We demonstrated the post-treatment PSA kinetics in prostate cancer patients treated with CIRT for the first time in a single-institution prospective observational study. PSA bounce can be distinguished from PSA failure in terms of timing of occurrence and amplitude, which will be useful information in the post-treatment surveillance of CIRT.

## Figures and Tables

**Figure 1 cancers-12-00589-f001:**
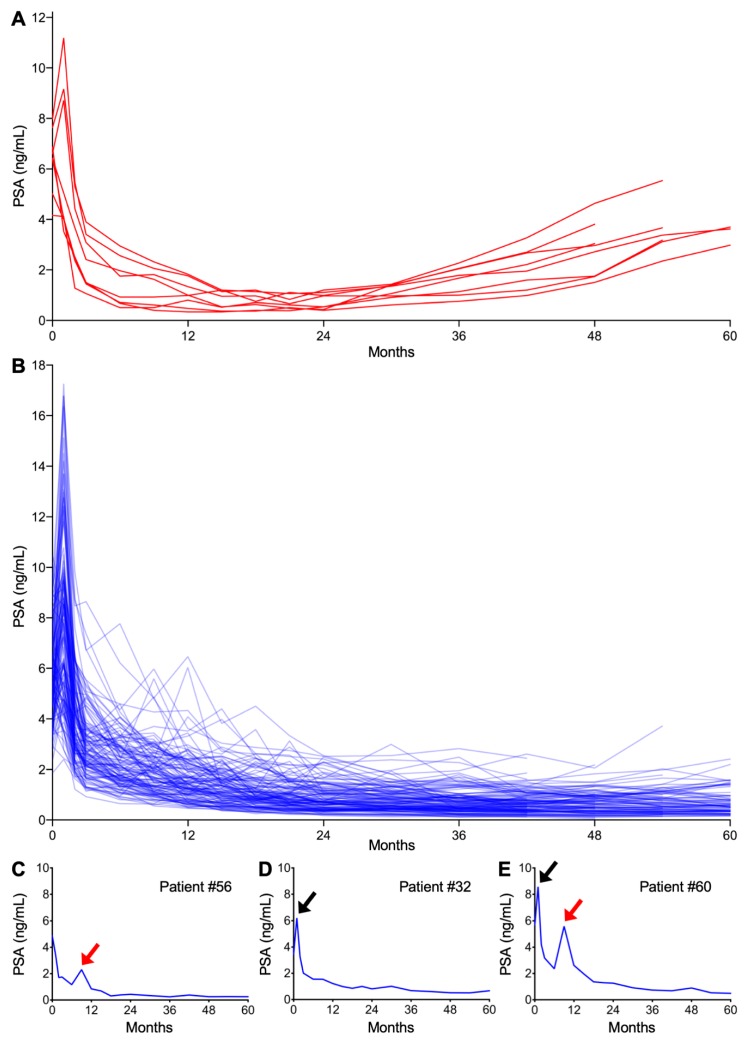
Post carbon ion radiotherapy (CIRT: carbon ion radiotherapy) prostate-specific antigen (PSA) kinetics. (**A**) PSA failure-positive patients (*n* = 8). (**B**) PSA failure-negative patients (*n* = 123). (**C**–**E**) Representative patients experiencing (**C**) PSA bounce, (**D**) surge, or (**E**) both. Red and black arrows indicate bounce and surge, respectively.

**Figure 2 cancers-12-00589-f002:**
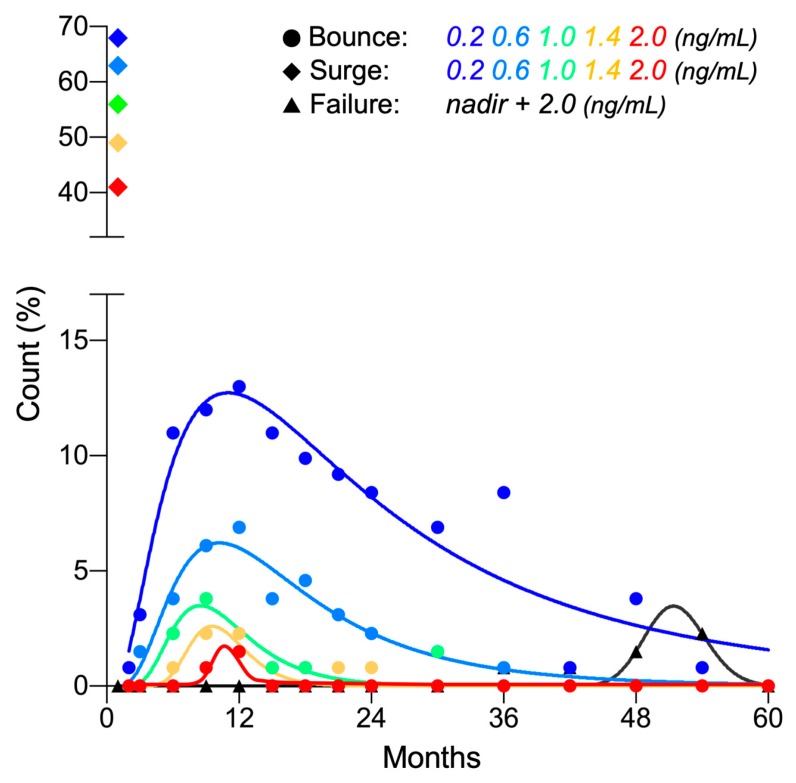
Frequency distribution of PSA bounce, surge, and failure post-CIRT (*n* = 131).

**Figure 3 cancers-12-00589-f003:**
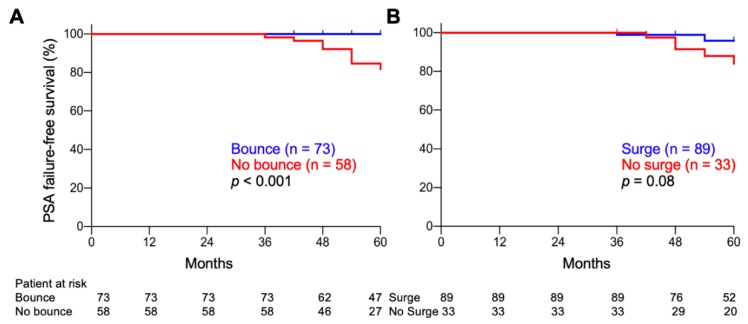
Kaplan–Meier estimates of PSA failure-free survival. (**A**) PSA bounce-positive vs. -negative patients. (**B**) PSA surge-positive vs. -negative patients. Cutoff, 0.2 ng/mL. *p*-values were determined by log-rank test.

**Table 1 cancers-12-00589-t001:** Patient characteristics.

Characteristics	All Patients (*n* = 131)
Age, mean (range)	64 (48–80)
T classification, *n* (%)	
T1c	60 (45.8)
T2a	63 (48.1)
T2b	8 (6.1)
Gleason score, *n* (%)	
6	30 (22.9)
7	101 (77.1)
Pretreatment PSA (ng/mL), median (range)	5.43 (1.86–10.47)
Prostate volume (mL), median (range)	37.82 (21.31–117.11)

PSA, prostate-specific antigen.

**Table 2 cancers-12-00589-t002:** Predictive factors for PSA bounce or surge.

Variables	PSA Bounce (≥0.2 ng/mL)						PSA Surge (≥0.2 ng/mL)					
	Univariate			Multivariate			Univariate			Multivariate		
	OR	95% CI	*p*	OR	95% CI	*p*	OR	95% CI	*p*	OR	95% CI	*p*
Age	0.92	0.87–0.97	0.003	0.92	0.87–0.98	0.008	0.97	0.91–1.03	0.31	0.96	0.90–1.02	0.21
T2a–T2b *	0.49	0.24–1.00	0.05	0.65	0.31–1.39	0.27	0.79	0.35–1.76	0.56	1.03	0.42–2.54	0.95
GS 7 **	0.80	0.35–1.83	0.59	0.97	0.40–2.36	0.95	0.37	0.12–1.17	0.09	0.47	0.14–1.61	0.23
Pretreatment PSA	0.88	0.73–1.07	0.20	0.88	0.71–1.07	0.20	0.87	0.70–1.08	0.21	0.82	0.65–1.04	0.11
Prostate volume	1.00	0.98–1.02	0.96	1.00	0.98–1.03	0.71	1.07	1.02–1.13	0.003	1.08	1.03–1.14	0.002

Results of logistic regression are shown. PSA, prostate-specific antigen; OR, odds ratio; CI, confidence interval; GS, Gleason score. * T1c as reference. ** GS 6 as reference. Bold words: statistically significant values. Bold words: statistically significant values.

**Table 3 cancers-12-00589-t003:** Predictive ability of PSA bounce or surge on PSA failure-free survival rate.

Cutoff (ng/mL)	PSA Bounce							PSA Surge						
	Bounce (+)			Bounce (−)				Surge (+)			Surge (−)			
	*n* at Risk	Rate	95% CI	*n* at Risk	Rate	95% CI	*p*	*n* at Risk	Rate	95% CI	*n* at Risk	Rate	95% CI	*p*
≥0.2	73	100	100–100	58	81.50	65.83–90.47	<0.001	89	95.79	87.36-98.64	33	84.76	64.04–94.05	0.08
≥0.6	33	100	100–100	98	89.57	80.01–94.70	0.09	82	95.37	86.16–98.51	40	87.49	69.69–95.18	0.19
≥1.0	17	100	100–100	114	91.03	82.66–95.46	0.27	75	94.94	84.94–98.36	47	89.21	73.45–95.86	0.34
≥1.4	9	100	100–100	122	91.57	83.68–95.75	0.43	64	94.25	83.04–98.13	58	91.05	77.58–96.60	0.61
≥2.0	3	100	100–100	128	91.93	84.34–95.93	0.62	54	95.57	83.09–98.90	68	90.64	78.76–96.03	0.45

Results of log-rank test are shown. PSA, prostate-specific antigen; CI, confidence interval. Bold words: statistically significant values.

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
