# Peer review of "Kinetics of Prostate-Specific Antigen after Carbon Ion Radiotherapy for Prostate Cancer"

_cancers, 2020, doi:10.3390/cancers12030589_

Round 1

Reviewer 1 Report

This paper examines for the first time PSA bounce after CIRT.  Their results were generated from 131 patients with prostate adenocarcinoma treated with CIRT. The results are reliable and interesting. The authors think their data are useful for post-CIRT surveillance of prostate cancer patients. However, they did not present the clinical implications/insight their results provide.

Author Response

We sincerely thank the reviewer for evaluating our manuscript. According to the suggestion, we added the clinical insight of the results in the Discussion section (lines 127131).

Reviewer 2 Report

This is a well written paper on post-treatment PSA kinetics in prostate cancer patients treated with CIRT. I was impressed with the quality of the data, as shown by standardized treatment, high follow-up compliance, and single institution PSA measurements. Importantly the ability to distinguish the PSA bounce from PSA failure is key for posttreatment surveillance of CIRT.

Authors discuss the data in a well organized manner and the data and discussion appears relevant for analysis of radiotherapy treatment on prostate cancer

Author Response

We sincerely thank the reviewer for evaluating our manuscript.

Reviewer 3 Report

The manuscript describes the PSA bounce and surge after CIRT, and their relationship toward PSA failure.

The manuscript is well-written and clear, and worth publishing in the journal.

Please check the format of references, make them consistent. Some are in title format, and some are not.

Author Response

We sincerely thank the reviewer for evaluating our manuscript. According to the suggestion, the format of references was thoroughly revised. We apologize for the inappropriate format.

Reviewer 4 Report

N. Darwis and team from Department of Radiation Oncology, Gunma University Graduate School of Medicine have performed the first PSA kinetic analysis of CIRT over period of 5 years. Their patients (131) were compliant in completing required blood draws for PSA measurement at >93% of study time points. The authors define PSA bounce from PSA surge. The authors compared the proportion of patients experiencing PSA bounce from CIRT to 18 other publications using other radiotherapy modalities. They found the quantified PSA bounce was greater using CIRT than any other modalities. The data collected confirms previous data (other radiotherapy modalities) showing younger patients have a high prevalence of PSA bounce and this is related to favorable PSA failure-free survival.  

Overall, the authors are defining baseline information to understand the relationship of PSA bounce and surge to CIRT, and then compare to other radiotherapies. CIRT causes dbl stand DNA breaks in cancer cells and a blood based marker (PSA) may help to determine CIRT efficacy versus other radiotherapies. The relationship between CIRT and the PSA bounce and surge  was demonstrated by Kaplan–Meier curve estimates of PSA failure-free survival in PSA bounce-positive vs. -negative patients p value <0.001. versus PSA surge-positive vs. -negative patients p value = 0.08. The detail biology of this observation remains to be rigorously determined.

I request the authors split Figure 1a into two figures. The original figure obscures limited PSA failure positive (RED data) patients versus failure negative (BLUE data) patients. The split figure will allow two sets of patients to be clearly viewed.

Author Response

We sincerely thank the reviewer for evaluating our manuscript. According to the suggestion, the original Figure 1A was separated into two (Figure 1A and 1B in the revised manuscript). We thank the reviewer for the insightful comment.
